# BrainSpace: a toolbox for the analysis of macroscale gradients in neuroimaging and connectomics datasets

Reinder Vos de Wael[1,7], Oualid Benkarim [1,7], Casey Paquola[1], Sara Lariviere[1], Jessica Royer [1], Shahin Tavakol[1], Ting Xu[1,2], Seok-Jun Hong [1,2], Georg Langs[3], Sofie Valk[4], Bratislav Misic [1], Michael Milham[2], Daniel Margulies[5], Jonathan Smallwood[6,7] & Boris C. Bernhardt[1,7 ✉]

Understanding how cognitive functions emerge from brain structure depends on quantifying how discrete regions are integrated within the broader cortical landscape. Recent work established that macroscale brain organization and function can be described in a compact manner with multivariate machine learning approaches that identify manifolds often described as cortical gradients. By quantifying topographic principles of macroscale organization, cortical gradients lend an analytical framework to study structural and functional brain organization across species, throughout development and aging, and its perturbations in disease. Here, we present BrainSpace, a Python/Matlab toolbox for (i) the identification of gradients, (ii) their alignment, and (iii) their visualization. Our toolbox furthermore allows for controlled association studies between gradients with other brain-level features, adjusted with respect to null models that account for spatial autocorrelation. Validation experiments demonstrate the usage and consistency of our tools for the analysis of functional and microstructural gradients across different spatial scales.

[1] McConnell Brain Imaging Centre, Montreal Neurological Institute and Hospital, McGill University, Montreal, Canada. [2] Center for the Developing Brain, Child Mind Institute, New York, USA. [3] Medical University of Vienna, Vienna, Austria. [4] Institute for Neuroscience and Medicine; 7/Institute of Systems Neuroscience, Forschungszentrum Juelich - Heinrich Heine Universitaet Duesseldorf, Juelich, Germany. [5] Frontlab, Institut du Cerveau et de la Moelle épinière, UPMC UMRS 1127, Inserm U 1127, CNRS UMR 7225, Paris, France. [6] Department of Psychology, University of York, Heslington, England, UK. [7] These authors contributed equally: Reinder Vos de Wael, Oualid Benkarim, Jonathan Smallwood, Boris C. Bernhardt. ✉email: boris.bernhardt@mcgill.ca

Over the last century, neuroanatomical studies in humans and non-human animals have highlighted two complementary features of neural organization. On the one hand, studies have demarcated structurally homogeneous areas with specific connectivity profiles, and ultimately distinct functional roles[1–5]. In parallel, neuroanatomists have established spatial trends that span across cortical areas both in terms of their histological properties, and connectivity patterns[6–9]. Such characterizations of cortical areas by their placement in the broader cortical hierarchy has provided a foundation for understanding functions that emerge through cortical interactions.

Although much of the more recent work linking measures of neural processing (for example from functional magnetic resonance imaging, MRI) to cognition has focused on identifying discrete regions and modules and their specific functional roles[10], recent conceptual and methodological developments have provided the data and methods that allow macroscale brain features mapped to low dimensional manifold representations, also described as gradients[11]. Gradient analyses operating on connectivity data were applied to diffusion MRI tractography data in specific brain regions[12,13] as well as neocortical, hippocampal, and cerebellar resting-state functional MRI connectivity maps[11,14–20]. Similar techniques have also been used to describe myelin-sensitive tissue measures and other morphological characteristics[21–23], as well as approaches based on combined network information aggregated from multiple features[24]. Other studies have used a similar framework to describe task based neural patterns either using meta-analytical co-activation mapping[20] or large-scale functional MRI task data sets[25]. Gradients have also been successfully derived from non-imaging data that were registered to stereotaxic space, including hippocampal post mortem gene expression information[26] and 3D histology data[22], to explore cellular and molecular signatures of neuroimaging and connectome measures. Core to these techniques is the computation of an affinity matrix that captures inter-area similarity of a given feature followed by the application of dimensionality reduction techniques to identify a gradual ordering of the input matrix in a lower dimensional manifold space (Fig. 1).

The ability to describe brain wide organizational principles in a single manifold offers the possibility to understand how the integrated nature of neural processing gives rise to function and dysfunction. Adopting a macroscale perspective on cortical organization has already provided insights into how cortex-wide patterns relate to cortical dynamics[27] and high level cognition[25,28–30]. Furthermore, several studies have leveraged gradients as an analytical framework to describe atypical macroscale brain organization across clinical conditions, for example, by showing perturbations in functional connectome gradients in autism[31] and schizophrenia[32]. Finally, comparisons of gradients across different imaging modalities have highlighted the extent to which structure directly constrains functional measures[22], while consideration of gradients across species has highlighted how evolution has shaped more integrative features of the cortical landscape[7,33–37].

The growth in our capacity to map whole brain cortical gradients, coupled with the promise of a better understanding of how structure gives rise to function, highlights the need for a set of tools that support the analysis of neural manifolds in a compact and reproducible manner. The goal of this paper is to present an open-access set of easy-to-use tools that allow the identification, visualization, and analysis of macroscale gradients of brain organization. We hope this will provide a method for calculating cortical manifolds that facilitates their use in future empirical work, allows comparison between studies, and allows for result replicability. To offer flexibility in implementation, we provide our toolbox in both Python and Matlab, two languages widely used in the neuroimaging and network neuroscience communities. Associated functions are freely available for download (http://github.com/MICA-MNI/BrainSpace) and complemented with an expandable online documentation (http://brainspace.readthedocs.io). We anticipate that our toolbox will assist researchers interested in studying gradients of cortical organization, and propel further work that establishes the overarching principles through which structural and functional organization of human and non-human brains gives rise to key aspects of cognition.

## Results

This section illustrates the usage of BrainSpace for gradient mapping and null model generation. Examples and evaluations are based on 217 subjects from the Human Connectome Project (HCP) dataset[38], as in prior work[20]. Matlab code is presented in the main version of the paper. Corresponding Python codes are available in the Supplementary Information.

**Generating gradients**. To illustrate the basic functionality of the toolbox, we computed gradients derived from resting-state functional MRI functional connectivity (FC). In short, the input matrix was made sparse (to 10% sparsity) and a cosine similarity matrix was computed. Next, three different manifold algorithms (i.e., principal component analysis (PCA), Laplacian eigenmaps (LE), diffusion mapping (DM)) were applied, followed by plotting their first and second gradients on the cortical surface (Figure 2). Resulting gradients (Fig. 3) derived from all dimensionality reduction techniques resemble those published previously[11], although for PCA the somatomotor to visual gradient explains more variance than the default mode to sensory gradient.

**Aligning gradients**. Gradient alignment across modalities. Based on subjects present in both the FC dataset as well as those used in the validation group of ref. [22] (n = 70), we examined the correspondence between gradients computed from different modalities and evaluated increases in correspondence through gradient

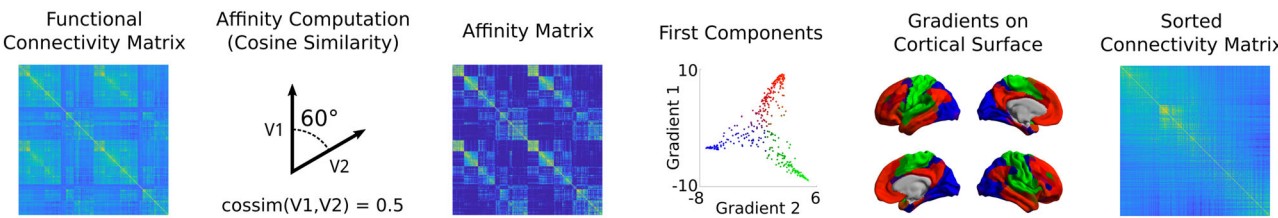

**Fig. 1 A typical gradient identification workflow.** Starting from an input matrix (here, functional connectivity), we use a kernel function to build the affinity matrix (here capturing the connectivity of each seed region). This matrix is decomposed, often via linear rotations or non-linear manifold learning techniques into a set of principal eigenvectors describing axes of largest variance. The scores of each seed onto the first two axes are shown in the scatter plot, with colors denoting position in this 2D space. These colors may be projected back to the cortical surface and the scores can be used to sort the input connectome.

alignment. The modalities evaluated are FC and microstructural profile covariance (MPC). Here we compared gradients of these in unaligned form, and after Procrustes alignment and joint embedding (Fig. 4, Fig. 5). As expected, gradient correspondence increased slightly following Procrustes alignment compared to unaligned gradients, and even more markedly following joint embedding. Beyond maximizing correspondence, the choice of Procrustes versus joint embedding can depend on the specific applications. Procrustes alignment preserves the overall shape of the different gradients and can thus be preferable to compare different gradients. Joint embedding, on the other hand, identifies a joint solution that maximizes their similarity, resulting in a gradient that may be more of a 'hybrid' of the input manifolds. Joint embedding is, thus, a technique to identify correspondence and to map from one space to another, and conceptually related to widely used multivariate associative techniques such as canonical correlation analysis or partial least squares which seek to maximize the linear associations between two multidimensional datasets[39]. Note that the computational cost of joint embedding is

```
% Create a GradientMaps object
G = GradientMaps('kernel','cosine', 'approach','dm');

% Apply GradientMaps to the data
G = G.fit(data_matrix);

% Load surfaces
left_surface = read_surface('left_surface_file.obj');
right_surface = read_surface('right_surface_file.obj');

% Plot first two gradients on the cortical surface
plot_hemispheres(G.gradients{1}(:,1:2), {left_surface,right_surface});
```

**Fig. 2 Sample code 1.** A minimal Matlab example for plotting the first gradient of an input data matrix on the cortical surface. Equivalent Python code is provided in Supplementary Sample Code 1.

substantially higher, so Procrustes analysis may be the preferred option when computational resources are a limiting factor.

Gradient alignment across individuals. Researchers may also be interested in comparing gradient values between individuals[31], for example to assess perturbations in FC gradients as a measure of brain network hierarchy. One possible approach could be to first build a group-level gradient template, to which both diagnostic groups are aligned using Procrustes rotation. After that, the two groups can be compared statistically and at each vertex, for example using tools suitable for surface-based linear modeling[40].

In the example below, we computed a template gradient from an out-of-sample dataset of 134 subjects from the HCP dataset (the validation cohort used by[20]). Next, we used the Procrustes analysis to align individual's gradients of each subject to the group level template (Fig. 6, Fig. 7).

Gradients across different spatial scales. The gradients presented so far were all derived at a vertex-wise level, which requires considerable computational resources. To minimize time and space requirements and to make results comparable to parcellation-based studies, some users may be interested in deriving gradients from parcellated data. To illustrate the effect of using different parcellations, we repeated the gradient identification and analysis across different spatial scales for both a structural and a functional parcellation. Specifically, we subdivided the conte69 surface into 400, 300, 200, and 100 parcels based on both a clustering of a well-established anatomical atlas[41], as well as a recently published local-global functional clustering[42] and built FC gradients from these representations (Fig. 8). These parcellations and subsampling schemes are provided in the shared folder of the BrainSpace toolbox.

Overall, with increasing spatial resolution the FC gradients became more pronounced and gradients derived from functional and structural parcellations were more similar. At a scale of 200 nodes or lower, putative functional boundaries may not be as reliably captured when using anatomically-informed parcellations,

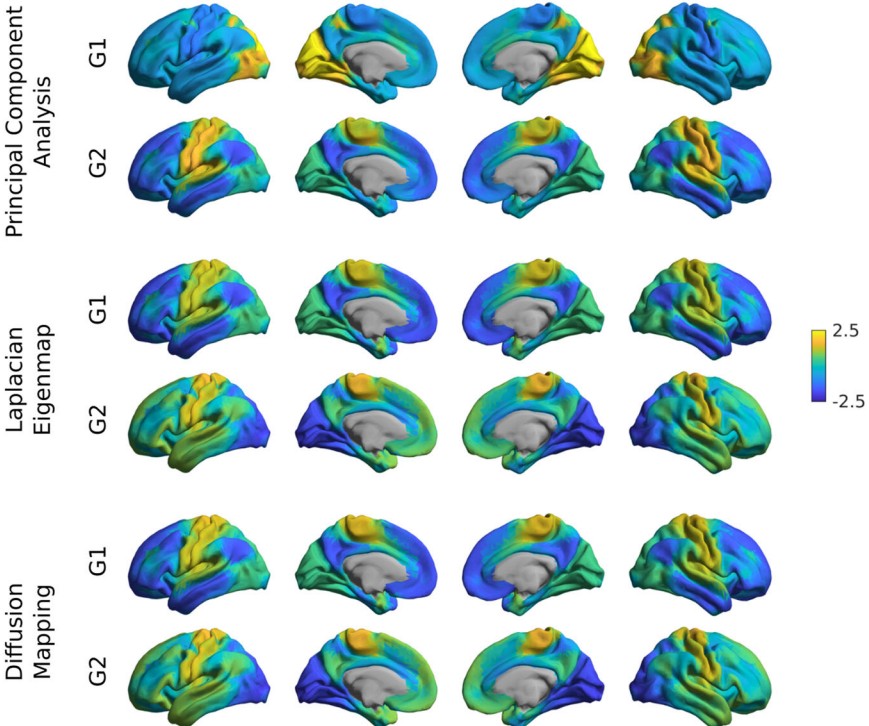

**Fig. 3 Gradient construction with different dimensionality reduction techniques.** Gradient 1 (G1) and 2 (G2) of FC were computed using a cosine similarity affinity computation, followed by either PCA, LE, or DM. Gradients were z-scored before plotting.

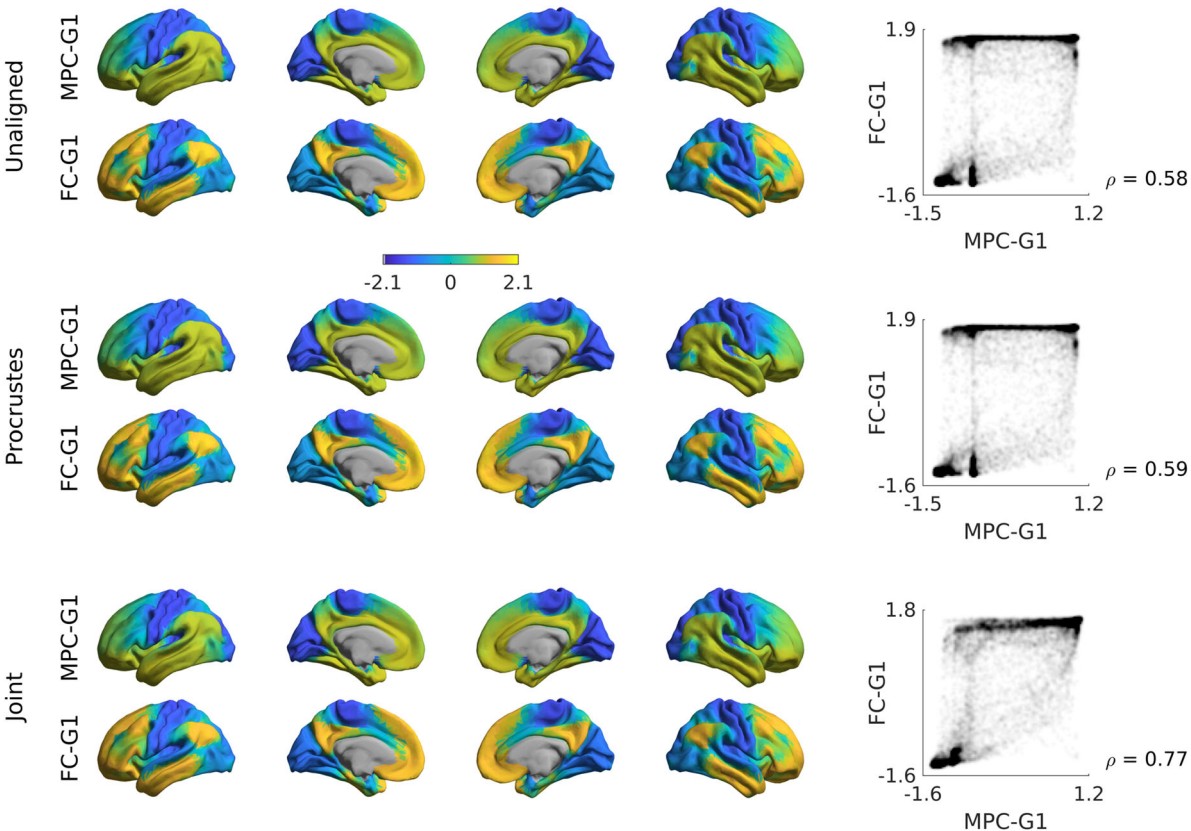

**Fig. 4 Comparison of alignment methods across modalities.** Unaligned gradients 1 (top) of MPC and FC were derived using cosine similarity and diffusion mapping. Alignments using Procrustes analyses (middle) and joint embedding (bottom) are also shown. Smoothed scatter plots show correspondence between principal gradient values for FC and MPC across cortical nodes, indicating a moderate increase in Spearman correlation after joint embedding. Gradient values were z-scored before plotting.

```matlab
% Create two GradientMaps objects with different alignments
Gp = GradientMaps('kernel','cosine', 'approach','dm', ...
                  'alignment','procrustes');
Gj = GradientMaps('kernel','cosine', 'approach','dm', ...
                  'alignment','joint');

% Apply GradientMaps to the data
Gp = Gp.fit({mpc,fc});
Gj = Gj.fit({mpc,fc}});

% Load surfaces
left_surface = read_surface('left_surface_file.obj');
right_surface = read_surface('right_surface_file.obj');

% Plot first MPC gradient of Procrustes alignment
plot_hemispheres(Gp.aligned{1}(:,1), {left_surface,right_surface});
```

**Fig. 5 Sample code 2.** A minimal Matlab example for creating and plotting gradients from different modalities, with different alignments. Equivalent Python code is provided in Supplementary Sample Code 2.

resulting in more marked alterations in the overall shape of the gradients.

For further evaluation, we related the above gradients across multiple scales relative to Mesulam's classic scheme of cortical laminar differentiation and hierarchy[22,43]. It shows a clear correspondence between the first gradient and the Mesulam hierarchy for high resolution data of 300 nodes and more, regardless of the parcellation scheme. While high correspondence was still seen for functional parcellations at lower granularity, it was markedly reduced when using structural parcellations. For

researchers interested in using Mesulam's parcellation, it has been provided on Conte69 surfaces in the shared folder of the BrainSpace toolbox. We also provide the data required to reproduce the evaluated FC and MPC gradients across these different spatial scales. Such gradients can be used to stratify other imaging measures, including functional activation and connectivity patters[28,31,44], meta-analytical syntheses[11,29], cortical thickness measures or Amyloid-beta PET uptake data[45].

**Null models**. Here, we present an example to assess the significance of correlations between FC gradients and data from other modalities (cortical thickness and T1w/T2w image intensity in this example). We present code (see Fig. 9) to generate previously proposed spin tests[46], which preserve the autocorrelation of the permuted features by rotating the feature data on the sphere. In our example (Fig. 10), the correlations between FC gradients and T1w/T2w stay significant (two-tailed, $p < 0.001$) even when comparing the correlation to 1000 null models whereas correlations between FC gradients and cortical thickness was non-significant (two-tailed, $p = 0.12$).

**Test-retest stability**. The HCP dataset has four rs-fMRI scans, split over two days. As such, we can leverage this data to assess test-retest stability of gradients. Here, we assessed the test-retest stability of gradients at the group level. Specifically, we redid the analysis of Fig. 2, but split the dataset by day of scanning. Stability was very high for LE and DM ($r > 0.99$) and moderate-to-high for PCA ($r \approx 0.72$) (Supplementary Fig. 1).

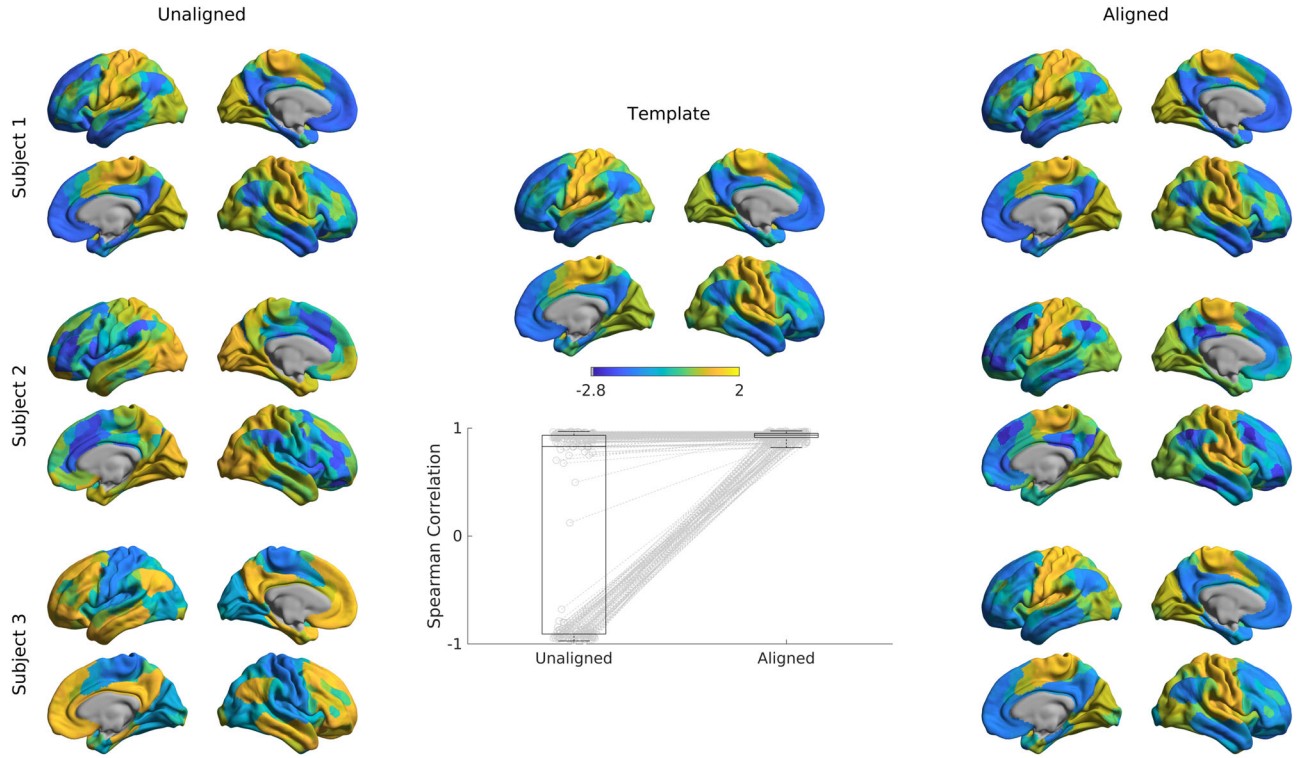

**Fig. 6 Alignment of subjects to a template.** Gradients 1 of single subjects computed with the cosine similarity kernel and diffusion mapping manifold (left) were aligned to an out-of-sample template (middle) using Procrustes analysis, creating aligned gradients (right). Box-plot shows the Spearman correlations of each subjects' gradient 1 values across cortical nodes to the template gradient 1 both before, and after Procrustes alignment. The central line of the box plot denotes the median, the edges denote the 25th and 75th percentiles, and the whiskers extend to the most extreme datapoints.

```matlab
% Create a GradientMaps object for the template
Gt = GradientMaps('kernel','cosine', 'approach','dm');

% Apply GradientMaps to template data
Gt = Gt.fit(template_data);

% Create a GradientMaps object for the individuals
Gs = GradientMaps('kernel','cosine', 'approach','dm', ...
                  'alignment','procrustes');

% Compute gradients of all subjects and align to template
Gs = Gs.fit({subject1_data,subject2_data}, ...
            'reference',Gt.gradients{1});

% Load surfaces
left_surface = read_surface('left_surface_file.obj');
right_surface = read_surface('right_surface_file.obj');

% Plot the first aligned gradient for subject 2
plot_hemispheres(Gs.aligned{2}(:,1), {left_surface,right_surface});
```

**Fig. 7 Sample code 3.** A minimal Matlab example for aligning the gradients of two individuals to a template gradient. Equivalent Python code is provided in Supplementary Sample Code 3.

## Discussion

While tools for unsupervised manifold identification and their alignment are extensively used in data science across multiple research domains[47], and while few prior connectome level studies made their workflow openly accessible (see refs. [11,15,48]), we currently lack a unified software package that incorporates the major steps of gradient construction and evaluation for neuroimaging and connectome datasets. We filled this gap with Brain-Space, a compact open-access Matlab/Python toolbox for the identification and analysis of low-dimensional gradients for any given regional or connectome-level feature. As such, BrainSpace

provides an entry point for researchers interested in studying gradients as windows into brain organization and function. BrainSpace is a simple and modular package accessible to beginners, yet expandable for advanced programmers. At its core is a simple object-oriented model, allowing for flexible computation of different (i) affinity matrices, (ii) dimensionality reduction techniques, (iii) alignment functions, and (iv) null models. We also supplied precomputed gradients, a novel subparcellation of the Desikan-Killiany atlas, and a literature-based atlas of cortical laminar differentiation that we used in a recent study[22,43].

As our main purpose was to provide an accessible introduction of the toolbox's basic functionality, we focused on tutorial examples and several selected assessments that demonstrate more general aspects of gradient analyses. First, relatively consistent FC gradients were produced by different dimensionality reduction techniques (i.e., PCA, LE, DM), at least when cosine similarity was chosen for affinity matrix computations. Interactions between input data, affinity matrix kernels, and dimensionality reduction techniques may nevertheless occur, a topic worthwhile to explore in future work. Second, we could show a relative consistency of FC gradients across spatial scales in the case of vertex-wise analyses and when parcellations with 300 nodes or more were used. However, we also observed an interaction between the type of input data and parcellation at lower spatial scales. In fact, lower resolution structural parcellations might not capture fine-grained functional boundaries, specifically in heteromodal and paralimbic association cortices which may be less constrained by underlying structural-morphological features[22]. It will be informative for future work to clarify how input modality (e.g., FC, MPC, or diffusion MRI), the choice of parcellation, and the spatial scale impact gradient analyses.

There are two broad ways through which the gradient method and the BrainSpace toolbox may improve our understanding of

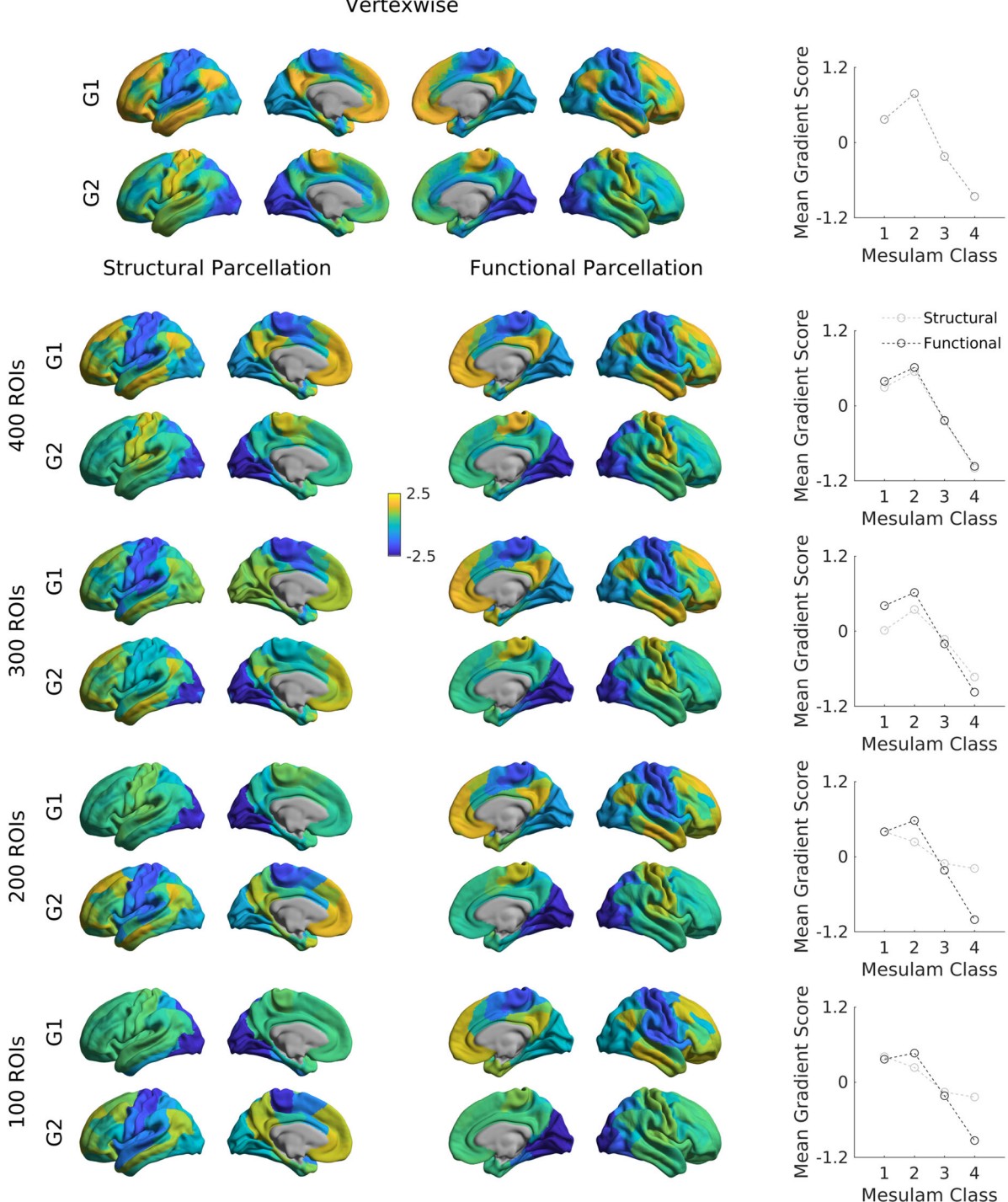

**Fig. 8 Functional gradients across spatial scales.** The cortex was subdivided into 100 (first row), 200 (second row), 300 (third row), and 400 (fourth row) regions of interest based on an anatomical (left) and functional (right) parcellation. Displayed are gradients 1 (G1), and 2 (G2), each for one hemisphere only. Line plots show the average gradient score within each Mesulam class for the functional (dark gray) and structural (light gray).

neural organization and its associated functions. One avenue is the identification of similarities and differences in gradients derived from different brain measures. To address associations between cortical microstructure and macroscale function, a previous study[22] demonstrated that gradients derived from 3D histology and myelin-sensitive MRI measures show both similarities and differences from those derived from resting state-state functional connectivity analysis[11]. This raises the possibility that the gradient method may help quantify common and distinct influences on

functional and structural brain organization and shed light on the neural basis of more flexible (i.e., less structurally constrained) aspects of human cognition[22]. Another way that gradients can inform our understanding of how functions emerge from the cortex is through the analysis of how macroscale patterns of organization change in disease. One recent study[31], for example, demonstrated differences in the principal functional gradient, identified by[11], between individuals with autism spectrum disorder and typically developing controls. In this way, manifold-

derived gradient analyses hold the possibility to characterize how macroscale functional organization may become dysfunctional in atypical neurodevelopment. To achieve both goals, alignment of different gradients should be considered, and this can be achieved by a number of methods we supplied, such as Procrustes alignment. This allows researchers to both compare gradients from different modalities and to homogenize measures across subjects, while minimizing the changes to individual manifolds. Our evaluations highlighted an increase in correspondence between individual subjects and the template manifold when Procrustes alignment was used compared to unaligned approaches, mainly driven by trivial changes in the sign of specific gradients in a subgroup of subjects. As an alternative to Procrustes alignment, it is also possible to align gradients via joint manifold alignments, often referred to as joint embedding[37]. The gradients provided by this approach can augment both cross-subject[49] as well as cross-species analyses[37]. Of note, joint embeddings generate a new manifold, which may result in new solutions that might slightly differ from the initial gradients computed individually.

When assessing the significance of correlations between gradients and other features of brain organization, there is an increasing awareness to ideally also evaluate correlations against null models with a similar spatial autocorrelation as the original features. BrainSpace provides two different approaches to build null models, including an adaptation of a previously released spin permutation test[46] and Moran's spectral randomization[50]. Gradients can also serve as a coordinate system, and stratify cortical features that are not gradient-based per se. Examples include surface-based geodesic distance measures from sensory-motor regions to other regions of cortex[11], task-based functional activation patterns and meta-analytical data[11,28,29], as well as MRI-based cortical thickness and PET-derived amyloid beta uptake measures[45]. As such, using manifolds as a new coordinate system[34] may complement widely used parcellation approaches[2,42,51] and be can be useful for the compact representation of findings and aid in the interpretation and communication of results.

```
% Load spheres
left_sphere = read_surface('left_sphere_file.obj');
right_sphere = read_surface('right_sphere_file.obj');

% Generate 1000 spin permutations
n_perm = 1000;
features_spin = spin_permutations({left_features,right_features}, ...
    {left_sphere,right_sphere},n_perm);
```

**Fig. 9 Sample code 4.** A minimal Matlab example for building null models based on spin tests. Equivalent Python code is provided in Supplemental Sample Code 4.

Although, we have only illustrated the use of our toolbox with neocortical surface data, the gradient-based functionality of BrainSpace is not restricted to these regions and can also be used with other datasets (e.g., hippocampal measures, subcortical measures, and volumetric data). However, as of version 0.1, visualization is only available for surface meshes. Extending the visualization functionality of BrainSpace to support other cerebral structures and volumetric data is a promising line of future work.

## Methods

**Input data description.** Our toolbox requires a real input matrix. Let $X \in \mathbb{R}^{n \times p}$ be a matrix aggregating features of several seed regions. In other words, each seed is represented by a $p$-dimensional vector, $x_i$, built based on the features of the $i$-th seed region, where $X(i,j) = x_i^j$ denotes the $j$-th feature of the $i$-th seed. In many neuroimaging applications, $X$ may represent a connectivity metric (e.g., resting-state functional MRI connectivity or diffusion MRI tractography derived structural connectivity) between different seed and target brain regions. When seed and target regions are identical, the input matrix $X$ is square. Furthermore, if the connectivity measure used to build the matrix is non-directional, $X$ is also symmetric. If seeds and targets are different, for example when assessing connectivity patterns of a given region with the rest of the brain[15,20], we may have that n ≠ p, resulting in a non-square matrix. The dimensions and symmetry properties of the input matrix $X$ may interact with the dimensionality reduction procedures presented in section 4.5. A simple strategy to make matrices symmetric and square is to use kernel functions, which will be covered in the following section.

**Affinities and kernel functions.** Since we are interested in studying the relationships between the seed regions in terms of their features (e.g., connectivity with target regions), our toolbox provides several kernel functions to compute the relationship between every pair of seed regions and derive a non-negative square symmetric affinity matrix $A \in {}^{n \times n}$, where $A(i,j) = A(j,i)$ denotes the similarity or 'affinity' between seeds $I$ and $j$. Moreover, a square symmetric matrix is a requirement for the next step in our framework (i.e., dimensionality reduction). Note that when the input matrix $X$ is already square and symmetric (e.g., seed and target regions are the same), there may be no need to derive the affinity matrix and $X$ can be used directly to perform dimensionality reduction. Accordingly, BrainSpace provides the option of skipping this step and using the input matrix as the affinity.

There are numerous kernels to compute affinity matrices. As of version 0.1, our toolbox implements the following: Gaussian, cosine similarity, normalized angle similarity, Pearson's correlation coefficient, and Spearman rank order correlations. For simplicity, let $x = x_i$ and $y = x_j$, these kernels can be expressed as follows:
1. Gaussian kernel:

$$A(i,j) = e^{-(\gamma \|x-y\|^2)},$$

where $\gamma$ is the inverse kernel width and $\|\bullet\|$ denotes the $l_2$-norm.
2. Cosine similarity:

$$A(i,j) = cossim(x,y) = \frac{xy^T}{\|x\|\|y\|},$$

where $cossim$ (•,•) is the cosine similarity function and $T$ stands for transpose.
3. Normalized angle similarity:

$$A(i,j) = 1 - \frac{\cos^{-1}(cossim(x,y))}{\pi}.$$

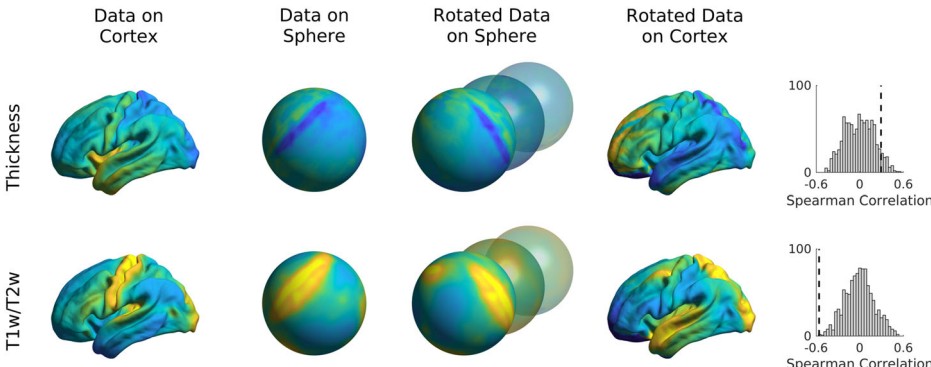

**Fig. 10 Spin tests of cortical thickness and t1w/t2w intensity.** Data were rotated on the sphere 1000 times and Spearman correlations between FC gradient 1 and the rotated data were computed. Distribution of correlation coefficients are shown in the histograms with the dashed lines denoting the true correlation.

4. Pearson correlation:

$$A(i,j) = \rho(\boldsymbol{x}, \boldsymbol{y}) = cossim(\boldsymbol{x} - \bar{\boldsymbol{x}}, \boldsymbol{y} - \bar{\boldsymbol{y}}),$$

where $\rho$ is the Pearson correlation coefficient, and $\bar{\boldsymbol{x}}$ and $\bar{\boldsymbol{y}}$ denote the means of $\boldsymbol{x}$ and $\boldsymbol{y}$, respectively.

5. Spearman rank order correlation:

$$A(i,j) = \rho\left(\boldsymbol{r_x}, \boldsymbol{r_y}\right),$$

where $\boldsymbol{r_x}$ and $\boldsymbol{r_y}$ denote the ranks of $\boldsymbol{x}$ and $\boldsymbol{y}$, respectively.

Version 0.1 of BrainSpace thus includes commonly used kernels in the gradient literature and additional ones for experimentation. To our knowledge, no gradient paper has used Pearson or Spearman correlation. Note that if $X$ is already row-wise demeaned, Pearson correlation amounts to cosine similarity. The Gaussian kernel is widely used in the machine learning community (for example in the context of Laplacian eigenmaps and support vector machines), which provides a simple approach to convert Euclidean distances between our seeds into similarities. Cosine similarity, (example application[11]:), computes the angle between our feature vectors to describe their similarity. Notably, cosine similarity ranges from -1 to 1, with negative correlations to be transformed to non-negative values. This motivated inclusion of the normalized angle kernel (example application:[20]), which circumvents negative similarities by transforming similarities to [0,1], with 1 denoting identical angles, and 0 opposing angles. Cosine similarity and Pearson and Spearman correlation coefficients may produce negative values (i.e., [−1,1]) if feature vectors are negatively correlated. A simple mitigation is to set negative values to zero[11]. Additionally, BrainSpace provides an option for row-wise thresholding of the input matrix[11,20,22,31]. Each vector of the input matrix is thresholded at a given sparsity (e.g., by keeping the weights of the top 10\% entries for each region). This procedure ensures that only strong, potentially less noisy, connections contribute to the manifold solution.

In addition to the aforementioned kernels, BrainSpace provides the option to provide a custom kernel or of skipping this step and using the input matrix as an affinity matrix.

**Dimensionality reduction**. In the input matrix, each seed $\boldsymbol{x}_i$ is defined by a $p$-dimensional feature vector, where $p$ may denote hundreds of parcels or thousands of vertices/voxels. The aim of dimensionality reduction techniques is to find a meaningful underlying low-dimensional representation, $\mathcal{G} \in \mathbb{R}^{n \times m}$ with $m \ll p$, hidden in the high-dimensional ambient space. These methods can be grouped into linear and non-linear techniques. The former use a linear transformation to unravel the latent representation, while techniques in the second category use non-linear transformations. As of version 0.1, BrainSpace provides three of the most widely used dimensionality reduction techniques for macroscale gradient mapping: PCA for linear embedding, and LE and DM for non-linear dimensionality reduction.

1. PCA is a linear approach that transforms the data to a low-dimensional space represented by a set of orthogonal components that explain maximal variance. Given a column-wise demeaned version of the input matrix $X_d$, the low-dimensional representation is computed as follows:

$$\mathcal{G}_{PCA} = US^T,$$

where $U$ are the left singular vectors and $S$ a diagonal matrix of singular values obtained after factorizing the input matrix using singular value decomposition, $X_d = USV^T$. Although here we present the singular value decomposition version below, PCA can also be performed via eigende-composition of the covariance matrix of $X$.

2. LE is a non-linear dimensionality reduction technique that uses the graph Laplacian of the affinity matrix $A$ to perform the embedding:

$$L = D - A,$$

where the degree matrix $D$ is a diagonal matrix defined as $D(i,i) = \sum_j A(i,j)$ and $L$ is the graph Laplacian matrix. Note that we can also work with its normalized version instead $L_S = D^{1/2}LD^{1/2}$[52]. LE then proceeds to solve the generalized eigenvalue problem:

$$L\boldsymbol{g} = \lambda D\boldsymbol{g},$$

where the eigenvectors $\mathbf{g}_k$ corresponding to the $m$ smallest eigenvalues $\lambda_k$ (excluding the first eigenvalue) are used to build the new low-dimensional representation:

$$\mathcal{G}_{LE} = [\boldsymbol{g}_1, \boldsymbol{g}_2, \dots, \boldsymbol{g}_m].$$

3. DM also seeks a non-linear mapping of the data based on the diffusion operator $P_\alpha$, which is defined as follows:

$$P_\alpha = D_\alpha^{-1} W_\alpha,$$

where $\alpha \in [0,1]$ is the anisotropic diffusion parameter used by the diffusion operator, $W_\alpha = D^{-1/\alpha} A D^{-1/\alpha}$ is built by normalizing the affinity matrix according

to the diffusion parameter and $D_\alpha$ is the degree matrix derived from $W_\alpha$. When $\alpha = 0$, the diffusion amounts to normalized graph Laplacian on isotropic weights, for $\alpha = 1$, it approximates the Laplace-Beltrami operator and for the case where $\alpha = 0.5$ it approximates the Fokker-Planck diffusion[53]. This parameter controls the influence of the density of sampling points on the manifold ($\alpha = 0$, maximal influence; $\alpha = 1$, no influence). In the gradient literature, the anisotropic diffusion hyper-parameter is commonly set to $\alpha = 0.5$[11,16,20], a choice that retains global relations between data points in the embedded space. Similar to LE, DM computes the eigenvalues and eigenvectors of the diffusion operator. However, in this case, the new representation is constructed with the scaled eigenvectors corresponding to the largest eigenvalues, after omitting the eigenvector with the largest eigenvalue:

$$\mathcal{G}_{DM} = [\lambda_1^t \boldsymbol{g}_1, \lambda_2^t \boldsymbol{g}_2, \dots, \lambda_m^t \boldsymbol{g}_m],$$

where $t$ is the time parameter that represent the scale.

All aforementioned dimensionality reduction approaches assume that our high-dimensional data lies on some low-dimensional manifold embedded in ambient space, which is typically the case with neuroimaging datasets. These techniques, therefore, provide a convenient approach to handle the curse of dimensionality inherent to neuroimaging data. Moreover, they facilitate comparison by recovering representations with the same number of dimensions even when source data is non-comparable in ambient space (for example when subjects have different fMRI time series of different lengths). PCA, in particular, is able to discover the low-dimensional structure when the data lies in an approximately linear manifold, but performs poorly when there are non-linear relationships within the data. In such scenarios, LE and DM are more appropriate to discover the intrinsic geometric structure. From a technical point of view, the advantage of PCA over the non-linear approaches included in BrainSpace is that it provides a mapping from the high- to the low-dimensional space rather than simply producing the new low-dimensional representations. Hence, the choice between linear and non-linear dimensionality reduction is problem-dependent and may also be influenced by the nature of the data under study.

**Alignment of gradients**. Gradients computed separately for two or more datasets (e.g., patients vs controls, left vs right hippocampi) may not be directly comparable due to different eigenvector orderings in case of eigenvalue multiplicity (i.e., eigenvalues with the same value) and sign ambiguity of the eigenvectors[54]. Aligning gradients improves comparability and correspondence. However, we recommend visually inspecting the alignment results; if the manifold spaces are substantially different, then alignments may not provide sensible output. In version 0.1 of the BrainSpace toolbox, gradients can be aligned using Procrustes analysis[55] or implicitly by joint embedding.

**Procrustes analysis**. Given a source $\mathcal{G}_s$ and a target $\mathcal{G}_t$ representation, Procrustes analysis seeks an orthogonal linear transformation $\psi$ to align the source to the target, such that $\psi(\mathcal{G}_s)$ and $\mathcal{G}_t$ are superimposed. Translation and scaling can also be performed by initially centering and normalizing the data prior to estimation of the transformation. For multiple datasets, a generalized Procrustes analysis can be employed. Let $\mathcal{G}_k, k = 1, 2, \dots, N$ be the low-dimensional representations of $N$ different datasets (i.e., input matrices $X_K$). The procedure proceeds iteratively by aligning all representations $\mathcal{G}_k$ to a reference and updating the reference $\mathcal{G}_R = \frac{1}{N}\sum_k \psi(\mathcal{G}_k)$ by averaging the aligned representations. In the first iteration, the reference can be chosen from the available representations, or an out-of-sample template can be provided (e.g., from a hold-out group).

**Joint embedding**. Joint embedding is a dimensionality reduction technique that finds a common underlying representation of multiple datasets via simultaneous embedding[37]. The main challenge of this technique is to find a meaningful approach to establish correspondences between original datasets (i.e., $X$). In version 0.1 of BrainSpace, joint alignment is implemented based on spectral embedding and available for LE and DM. The only difference with these methods is that the embedding, rather than using the affinity matrices individually, is based on the joint affinity matrix $\mathcal{J}$, built as:

$$\mathcal{J} = \begin{pmatrix} A_1 & A_{12} & \cdots & A_{1N} \\ A_{12}^T & A_2 & \cdots & A_{2N} \\ \vdots & \vdots & \ddots & \vdots \\ A_{1N}^T & A_{2N}^T & \cdots & A_N \end{pmatrix}$$

where $A_k$ is the intra-dataset affinity of the input matrix $X_k$ and $A_{ij}$ is the inter-dataset affinity between $X_i$ and $X_j$. As of version 0.1, both sets of affinities are built using the same kernel. It is important to note, therefore, that joint embedding can only be used if the input matrices have the same features (e.g., identical target regions). After the embedding, the resulting shared representation $\mathcal{G}_J = [\mathcal{G}_1, \mathcal{G}_2, \dots, \mathcal{G}_N]^T$ will be composed of $N$ individual low-dimensional representations, such that for the k-th input matrix $X_k \in \mathbb{R}^{n_k \times m}$, the corresponding representation is $\mathcal{G}_k \in \mathbb{R}^{n_k \times m}$, where $n_k$ is the number of seeds (i.e., rows) of $X_k$.

**Null models**. Many researchers have compared gradients to other continuous brain markers such as cortical thickness measures or estimates of cortical myelination. Given the spatial autocorrelation present in many modalities, applying linear regression or similar methods may provide biased test statistics. To circumvent this issue, we recommend comparing the observed test statistic to those of a set of distributions with similar spatial autocorrelation. To this end, we provide two methods: spin permutations[46] and Moran spectral randomization (MSR)[56,57]. In cases where the input data lies on a surface and most of the sphere is used or if data can be mapped to a sphere, we recommend spin permutation. Otherwise, we recommend MSR. When performing a statistical test with multiple gradients as either predictor or response variable, we recommend randomizing the non-gradient variable as these randomizations need not maintain statistical independence across different eigenvectors.

**Spin permutations**. Spin permutation analysis leverages spherical representations of the cerebral cortex, such as those derived from FreeSurfer[58] or CIVET[59], to address the problem of spatial autocorrelation in statistical inference. Spin permutations estimate the null distribution by randomly rotating the spherical projections of the cortical surface while preserving the spatial relationships within the data[46]. Let $V_r \in \mathbb{R}^{l \times 3}$ be the matrix of vertex coordinates in the sphere, where $l$ is the number of vertices, and $R \in \mathbb{R}^{3 \times 3}$ a matrix representing a rotation along the three axes uniformly sampled from all possible rotations[60]. The rotated sphere $V_r$ is computed as follows:

$$V_r = VR.$$

Samples of the null distribution are then created by assigning each vertex on $V_r$ the data of its nearest neighbor on $V$.

**Moran spectral randomization**. Borrowed from the ecology literature, MSR can also be used to generate random variables with identical or similar spatial auto-correlation (in terms of Moran's $I$ correlation coefficient[50]). This approach requires building a spatial weight matrix defining the relationships between the different locations. For our particular case, given a surface mesh with $l$ vertices (i.e., locations), its topological information is used to build the spatial weight matrix $L \in \mathbb{R}^{l \times l}$, such that $L(i, j) > 0$ if vertices $i$ and $j$ are neighbors, and $L(i, j) = 0$ otherwise. In BrainSpace, $L$ is built using the inverse distance between each vertex and the vertices in its immediate neighborhood, although other neighborhoods and weighting schemes (e.g., binary or Gaussian weights) can also be incorporated. The computed $L$ is doubly centered and eigendecomposed into its whole spectrum, with the resulting eigenvectors $M \in \mathbb{R}^{l \times l-1}$ being the so-called Moran eigenvector maps. Note that eigenvectors with 0 eigenvalue are dropped. One advantage of MSR is that we can work with the original cortical surfaces, and thus skip potential distortions introduced by spherical mesh parameterization.

To generate null distributions, let $u \in \mathbb{R}^l$ be an input feature vector defined on each vertex of our surface (e.g., cortical thickness), and $r \in \mathbb{R}^{l-1}$ the correlation coefficients of $u$ with each spatial eigenvector in $M$. MSR aims to find a randomized feature vector $z$ that respects the autocorrelation observed in $u$ as follows:

$$z = \bar{u} + \sigma_x \sqrt{l-1} \mathcal{M} a^T,$$

where $\bar{u}$ and $\sigma_u$ stand for mean and standard deviation of $u$ respectively, and $a \in \mathbb{R}^{l-1}$ is a vector of random coefficients. Three different methods exist for generating vector $a$: singleton, pair, and triplet. As of version 0.1, only the singleton and pair procedures are supported in BrainSpace. Let $v = Mr^T$, in the singleton procedure, $a$ is computed by randomizing the sign of each element in $v$ (i.e., $a_i = \pm v_i$). In the pair procedure, the elements of $v$ are randomly changed in pairs. Let $(v_i, v_j)$ be a pair of elements randomly chosen, then $a$ is updated such that $a_i = q_{ij} \cos(\phi)$ and $a_j = q_{ij} \sin(\phi)$, with $q_{ij} = \sqrt{v_i^2 + v_j^2}$ and $\phi \sim \mathcal{U}(0, 2\pi)$ randomly drawn from a uniform distribution. If the number of elements of $v$ is odd, the singleton procedure is used for the remaining element.

As opposed to the singleton procedure, the null data generated by the pair procedure does not fully preserve the observed spatial autocorrelation. We therefore recommend the singleton procedure, unless the number of required randomizations exceeds $2^{l-1}$, which is the maximum number of unique randomizations that can be produced using the singleton procedure.

**Participants**. All data were derived from the Human Connectome Project dataset[38]. For all figures showing results based only on FC, we included subjects processed for a prior study ($n = 217$ (122 women), mean $\pm$ SD age $= 28.5 \pm 3.7$ y)[20]. In short, these were subjects for whom all resting-state fMRI and anatomical scans were fully completed, and no familial relationships existed between subjects in this study. Data shown for comparisons between FC and MPC were derived from the subjects available both in the prior dataset as well as the data used by ref. [22] [$n = 70$ (41 women), mean $\pm$ SD age $= 28.7 \pm 3.9$ y]. As part of the Human Connectome Project acquisitions, informed consent was obtained from all subjects and all procedures were approved by the Washington University institutional review board.

**Reporting summary**. Further information on research design is available in the Nature Research Reporting Summary linked to this article.

## Data availability

All data are freely provided by the Human Connectome Project[38] and available from connectomeDB[61] (https://db.humanconnectome.org/).

## Code availability

Our toolbox is freely available at: http://github.com/MICA-MNI/BrainSpace. The toolbox contains a parallel Python and Matlab implementation with closely-matched functionality (The Python implementation of BrainSpace incorporates a wrapper for VTK, which helps simplify object creation and pipelining). Along with the code, the toolbox contains several surface models, parcellations across multiple scales, and example data to reproduce the evaluations presented in this tutorial. Additional documentation of the proposed toolbox is available for both Python and Matlab implementations via http://brainspace.readthedocs.io.

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

## Acknowledgements

Mr. Vos de Wael was funded by the Savoy Foundation for Epilepsy Research. Dr. Oualid Benkarim was funded by a Healthy Brains for Healthy Lives (HBHL) postdoctoral fellowship. Dr. Jessica Royer was supported by a Canadian Open Neuroscience Platform (CONP) fellowship. Dr Smallwood was supported by the European Research Council (WANDERINGMINDS—ERC646927). Dr. Paquola was funded through a postdoctoral fellowship of the Transforming Autism Care Consortium (TACC) and the Fonds de la recherche du Québec—Santé (FRQ-S). Dr. Boris Bernhardt acknowledges research support from the National Science and Engineering Research Council of Canada (NSERC, Discovery-1304413), the Canadian Institutes of Health Research (CIHR, FDN-154298), the Azrieli Center for Autism Research of the Montreal Neurological Institute (ACAR), SickKids Foundation (NI17-039), and the Canada Research Chairs (CRC) Program. Data were provided, in part, by the Human Connectome Project, WU-Minn Consortium (Principal Investigators: David Van Essen and Kamil Ugurbil; 1U54MH091657) funded by the 16 NIH Institutes and Centers that support the NIH Blueprint for Neuroscience Research; and by the McDonnell Center for Systems Neuroscience at Washington University.

## Author contributions

R.V., O.B., J.S. and B.B. developed the toolbox, designed the validations, and wrote the paper. S.L., D.M., J.R., S.T., S.V, C.P. and S.H., tested the toolbox and revised the paper. M.M., G.L., T.X. and B.M. revised and approved the paper.

## Competing interests

The authors declare no competing interests.
