## [Peer Review File · Communications Biology]

REVIEWERS' COMMENTS:

Reviewer #1 (Remarks to the Author):

Vos de Wael and colleagues present an extraordinary and unprecedented set of tools that has the potential to greatly accelerate and improve the quality of gradient-based neuroimaging analyses. I agree with the authors' claims that BrainSpace will make gradient analyses more accessible to the neuroimaging community, contribute to the important mission of open and reproducible neuroscience, and enhance the quality of the analyses in this new and expanding branch of neuroimaging research.

I was able to download their software and successfully run all the examples that are presented in the manuscript.

I fully support the publication of this manuscript. I believe the following minor recommendations might improve the quality of this excellent paper:

1 - Readers that are not focused on cerebral cortical science might benefit from an explanation of what aspects of BrainSpace (if any) are designed to work exclusively on cerebral cortical data. My understanding is that all functions except the visualization of results should work, but it may be useful to say this explicitly in the manuscript since all examples that are presented analyze only cerebral cortical data. It might also be helpful to mention how subcortical results, or volumetric cerebral cortical results, could be visualized (for example by mentioning what neuroimaging visualization matlab and python toolbox should be able to open and view subcortical / cerebral cortical volumetric results calculated with BrainSpace).

2 - The authors provide very useful information discussing cases when particular methods of affinity matrix calculation might be better than others. The same is true for methods of alignment. I wonder whether it would be possible to provide a short discussion of cases when specific methods of dimensionality reduction (PCA, LE, DM) might be better than others, or highlight some advantages/disadvantages of each method.

3 - The authors make reference to the "current" version of BrainSpace on multiple occasions in the manuscript. It would be useful to state what this current version is, so that readers have this as a reference when a new version of BrainSpace becomes available.

4 - It might be helpful to mention either in the manuscript or in "readthedocs.io" what systems has this software been tested in. It is extremely valuable that the authors have made the toolbox available in both Matlab and Python, offering the community a free (Python) alternative. It may also be helpful to mention if the authors have tested their software with free alternatives to Matlab such as Octave (if not, the authors might simply state, for example, that the software has not been tested with Octave). Similarly, it would be useful to mention if the software has been tested running Jupyter notebooks within the Ubuntu app that is available in the last versions of Windows (I tried this and could run code but not visualize figures, while I could both run code and visualize figures when using Matlab, but this might be something specific of my computer). A simple list of the systems that the authors have tried would be valuable.

5 - Page 6, line 145, "solutino" should be "solution". Page 14, line 353, "become" should be "became" (as the following verb in the same sentence, "were", is written in past tense).

6 - This is a very minor recommendation, but I believe referencing this paper might be relevant when reviewing prior gradient-based literature, since I believe this is the only gradient-based analysis of cerebellar cortex: "Functional gradients of the cerebellum", <https://elifesciences.org/articles/36652>

Thank you for the opportunity to review this manuscript.

Xavier Guell

Reviewer #2 (Remarks to the Author):

The authors present "BrainSpace", a software framework for macroscale connectivity analysis of neuroimaging datasets.

The MAIN CONTRIBUTION of this work is the open and freely available software toolkit consisting of extremely well-documented source code. In addition to the toolkit, the manuscript offers a rather compact (in a good way!) summary of existing algorithms and analysis methods. The authors succeed in providing a useful compendium and assembly of existing knowledge with up-to-date references for further information.

The manuscript itself is very well-written. All illustrations are clean and self-contained.

* Introduction: This is nicely done, and the research story is complete.

* Methodology: Again, a great compact summary of existing algorithms and analysis methods. Some repetition could be removed, for instance, lines 75-81 are similar to lines 395-400. All formulas look correct and well described. The contribution here is the compressed summary of existing knowledge in the field (very interesting and useful but no novel insights).

* Examples and evaluations: The authors describe four different use-cases and examples where "BrainSpace" can support active research. All examples are accompanied with Matlab and Python code. While very applied and useful, this is a rather qualitative evaluation. Around lines 450, the authors hint at scientific findings, but details are missing and the claims are not backed up. The figures in this section are beautiful but not scientifically analyzed. All examples are fully reproducible!

* Appendix: Nice and compact.

SOFTWARE ENGINEERING (MAIN CONTRIBUTION):

The source code of "BrainSpace" is publicly available, which is great. The source code is also extremely well documented with rigorous comments for every functionality (impressive!). I was also very fascinated by the available extensive web-based documentation that includes many examples for both Matlab and Python. For me, a handy feature of "BrainSpace" is the `serial_connect` functionality for vtk development. This functionality allows to simplify the typical vtk-style chaining of processing pipelines from multiple blocks of code to just a few lines (https://brainspace.readthedocs.io/en/latest/generated/brainspace.vtk_interface.pipeline.serial_connect.html). This is something I suggest to mention in the manuscript as well.

One question I had when reading the paper was, "Can BrainSpace easily create the beautiful 2D and 3D plots of the paper?". I was able to answer this question using the web-based documentation and see how to do it. Thank you.

In general, I would like to suggest to use Python as the language for all examples in the paper since it better fits the open science community. And, this community effort of "BrainSpace" is excellent: The authors request for papers to be added to the documentation, use Github for collaborative coding, and

leverage the Python/PIP ecosystem for easy deployment and user-friendly installation.

The main point for a POSSIBLE EXPANSION of this work is the LACK OF A FORMAL EVALUATION. The authors state that the manuscript focuses on tutorials rather than scientific insights (line 405). However, including a quantitative evaluation would allow the authors to support the usefulness of their contribution fully.

1 Reviewer #1

Vos de Wael and colleagues present an extraordinary and unprecedented set of tools that has the potential to greatly accelerate and improve the quality of gradient-based neuroimaging analyses. I agree with the authors' claims that BrainSpace will make gradient analyses more accessible to the neuroimaging community, contribute to the important mission of open and reproducible neuroscience, and enhance the quality of the analyses in this new and expanding branch of neuroimaging research.

I was able to download their software and successfully run all the examples that are presented in the manuscript.

I fully support the publication of this manuscript. I believe the following minor recommendations might improve the quality of this excellent paper:

We are delighted that the Reviewer is so positive about our manuscript, and are thankful for the constructive comments and suggestions.

R1.1. Readers that are not focused on cerebral cortical science might benefit from an explanation of what aspects of BrainSpace (if any) are designed to work exclusively on cerebral cortical data. My understanding is that all functions except the visualization of results should work, but it may be useful to say this explicitly in the manuscript since all examples that are presented analyze only cerebral cortical data. It might also be helpful to mention how subcortical results, or volumetric cerebral cortical results, could be visualized (for example by mentioning what neuroimaging visualization matlab and python toolbox should be able to open and view subcortical / cerebral cortical volumetric results calculated with BrainSpace).

As the Reviewer suggests, all code but the surface visualization functionality should work irrespective of whether the data was derived from a (subcortical) volume or surface. We have clarified this in the discussion section:

Although, we have only illustrated the use of our toolbox with neocortical surface data, the gradient-based functionality of BrainSpace is not restricted to these regions and can also be used with other datasets (e.g., hippocampal measures, subcortical measures, and volumetric data). However, as of version 0.1, visualization is only available for surface meshes. Extending the visualization functionality of BrainSpace to support other cerebral structures and volumetric data is a promising line of future work.

R1.2. The authors provide very useful information discussing cases when particular methods of affinity matrix calculation might be better than others. The same is true for methods of alignment. I wonder whether it would be possible to provide a short discussion of cases when specific methods of dimensionality reduction (PCA, LE, DM) might be better than others, or highlight some advantages/disadvantages of each method.

Indeed, our manuscript lacked a discussion on these differences. PCA distinguishes itself from the others by being the only linear method. As such, it is more interpretable but unable to describe non-linear patterns in the data. If

non-linear geometry exists within the data, then LE and DM are more suitable. A more detailed discussion has been added to the revised manuscript and is included here for convenience:

All aforementioned dimensionality reduction approaches assume that our high-dimensional data lies on some low-dimensional manifold embedded in ambient space, which is typically the case with neuroimaging datasets. These techniques, therefore, provide a convenient approach to handle the curse of dimensionality inherent to neuroimaging data. Moreover, they facilitate comparison by recovering representations with the same number of dimensions even when source data is non-comparable in ambient space (for example when subjects have different fMRI time series of different lengths). PCA, in particular, is able to discover the low-dimensional structure when the data lies in an approximately linear manifold, but performs poorly when there are non-linear relationships within the data. In such scenarios, LE and DM are more appropriate to discover the intrinsic geometric structure. From a technical point of view, the advantage of PCA over the non-linear approaches included in BrainSpace is that it provides a mapping from the high- to the low-dimensional space rather than simply producing the new low-dimensional representations. Hence, the choice between linear and non-linear dimensionality reduction is problem-dependent and may also be influenced by the nature of the data under study.

R1.3. The authors make reference to the “current” version of BrainSpace on multiple occasions in the manuscript. It would be useful to state what this current version is, so that readers have this as a reference when a new version of BrainSpace becomes available.

This is an excellent suggestion. The current release on Github is version 0.1. As such, we have changed all statements of “current version” or similar language to “version 0.1”.

R1.4. It might be helpful to mention either in the manuscript or in “readthedocs.io” what systems has this software been tested in. It is extremely valuable that the authors have made the toolbox available in both Matlab and Python, offering the community a free (Python) alternative. It may also be helpful to mention if the authors have tested their software with free alternatives to Matlab such as Octave (if not, the authors might simply state, for example, that the software has not been tested with Octave). Similarly, it would be useful to mention if the software has been tested running Jupyter notebooks within the Ubuntu app that is available in the last versions of Windows (I tried this and could run code but not visualize figures, while I could both run code and visualize figures when using Matlab, but this might be something specific of my computer). A simple list of the systems that the authors have tried would be valuable.

Both Matlab and Python implementations of BrainSpace have been tested on macOS Mojave and Linux Ubuntu 16.04. Moreover, for the Python package, we have tested the code in several system configurations using the *pytest* testing framework and continuous integration through *Travis CI* (Ubuntu Xenial 16.04, and macOS 10.13 and 10.14) and *AppVeyor* (Windows Server 2016 and 2019). In accordance with the Reviewer’s suggestion, we will update the installation

guide of our readthedocs with this list of systems. However, in order to maintain consistency between the two implementations, Github, readthedocs and PyPI, we would prefer to include this information with the soon-to-be-released version 0.2 of BrainSpace.

As of version 0.1, BrainSpace is not Octave compatible as it uses functionality exclusive to Matlab and, unfortunately, our future plans do not include making it compatible. Regarding the Python visualization problem with the Ubuntu app in Windows, it might be a problem with VTK, although no tests have been run in this environment. If interested, we invite the Reviewer to open an issue on Github and describe the problem in order to find a possible solution.

R1.5. Page 6, line 145, “solutino” should be “solution”. Page 14, line 353, “become” should be “became” (as the following verb in the same sentence, “were”, is written in past tense).

We thank the Reviewer for catching these typographical errors. They have been amended accordingly in the revised manuscript.

R1.6. This is a very minor recommendation, but I believe referencing this paper might be relevant when reviewing prior gradient-based literature, since I believe this is the only gradient-based analysis of cerebellar cortex: “Functional gradients of the cerebellum”, <https://elifesciences.org/articles/36652>.

We agree that adding this reference would improve our introduction of the literature, and have added it to the manuscript. The new introduction now states:

Gradient analyses operating on connectivity data were applied to diffusion MRI tractography data in specific brain regions (Cerliani et al., 2012; Bajada et al., 2017) as well as neocortical, hippocampal, and cerebellar resting-state functional MRI connectivity maps (Margulies et al., 2016; Vos de Wael et al., 2018; Lariviere et al., 2019; Tian and Zalesky, 2018; Marquand et al., 2017; Haak et al., 2018; Przewdzik et al., 2019; Guell et al., 2018).

R1.7. Thank you for the opportunity to review this manuscript. Xavier Guell We thank Dr Guell for the thoughtful review and positive evaluation

2 Reviewer #2

The authors present “BrainSpace”, a software framework for macroscale connectivity analysis of neuroimaging datasets.

The MAIN CONTRIBUTION of this work is the open and freely available software toolkit consisting of extremely well-documented source code. In addition to the toolkit, the manuscript offers a rather compact (in a good way!) summary of existing algorithms and analysis methods. The authors succeed in providing a useful compendium and assembly of existing knowledge with up-to-date references for further information.

The manuscript itself is very well-written. All illustrations are clean and self-contained.

We thank the Reviewer for the positive evaluation and helpful comments.

R2.1 Methodology: Again, a great compact summary of existing algorithms and analysis methods. Some repetition could be removed, for instance, lines 75-81 are similar to lines 395-400. All formulas look correct and well described. The contribution here is the compressed summary of existing knowledge in the field (very interesting and useful but no novel insights).

The Reviewer is right that these lines are a bit repetitive. We have removed references to the documentation and github from the discussion (formerly lines 395-400).

We filled this gap with BrainSpace, a compact open-access Matlab/Python toolbox for the identification and analysis of low-dimensional gradients for any given regional or connectome-level feature. As such, BrainSpace provides an entry point for researchers interested in studying gradients as windows into brain organization and function.

R2.2 Examples and evaluations: The authors describe four different use-cases and examples where “BrainSpace” can support active research. All examples are accompanied with Matlab and Python code. While very applied and useful, this is a rather qualitative evaluation. Around lines 450, the authors hint at scientific findings, but details are missing and the claims are not backed up. The figures in this section are beautiful but not scientifically analyzed. All examples are fully reproducible!

We thank the Reviewer for noticing that our figure analysis could be improved. As this comment is similar to comment R2.4, we refer the Reviewer to our answer there.

SOFTWARE ENGINEERING (MAIN CONTRIBUTION):

The source code of “BrainSpace” is publicly available, which is great. The source code is also extremely well documented with rigorous comments for every functionality (impressive!). I was also very fascinated by the available extensive web-based documentation that includes many examples for both Matlab and Python. For me, a handy feature of “BrainSpace” is the `serial_connect` functionality for `vtk` development. This functionality allows to simplify the typical `vtk`-style chaining of processing pipelines from multiple blocks of code to just a few lines (https://brainspace.readthedocs.io/en/latest/generated/brainspace.vtk_interface.pipeline.serial_connect.html). This is something I suggest to mention in the manuscript as well.

One question I had when reading the paper was, “Can BrainSpace easily create the beautiful 2D and 3D plots of the paper?”. I was able to answer this question using the web-based documentation and see how to do it. Thank you.

We are glad the Reviewer navigated our documentation with such ease. Following the Reviewer’s suggestion, the revised manuscript has been updated with a footnote mentioning the functionality provided by the VTK wrapper implemented in Python:

The Python implementation of BrainSpace incorporates a wrapper for VTK, which helps simplify object creation and VTK's pipelining.

R2.3 In general, I would like to suggest to use Python as the language for all examples in the paper since it better fits the open science community. And, this community effort of "BrainSpace" is excellent: The authors request for papers to be added to the documentation, use Github for collaborative coding, and leverage the Python/PIP ecosystem for easy deployment and user-friendly installation.

We are happy that the Reviewer appreciates our open-science efforts. The main reason we used Matlab in the manuscript is the fact that all figures in our manuscript were created with the Matlab version of BrainSpace, and we would prefer to keep this consistency. Nevertheless, each Matlab sample code in our manuscript also provides a reference to its analogous Python implementation.

R2.4 The main point for a POSSIBLE EXPANSION of this work is the LACK OF A FORMAL EVALUATION. The authors state that the manuscript focuses on tutorials rather than scientific insights (line 405). However, including a quantitative evaluation would allow the authors to support the usefulness of their contribution fully.

The Reviewer is correct that our results section is, largely, qualitative. We structured our manuscript this way to keep the focus on how to construct and interpret gradients. Nevertheless, we agree that some quantitative analysis may improve the manuscript. To that end we assessed test-retest stability. The HCP dataset provides four resting-state scans taken across two days. We compared gradients produced by scans from the first day vs those from the second day. This resulted in a new sub-section and supplementary figure.

To further distinguish our quantitative and qualitative claims, we have also included exact p-values in the null model section.

The HCP dataset has four rs-fMRI scans, split over two days. As such, we can leverage this data to assess test-retest stability of gradients. Here, we assessed the test-retest stability of gradients at the group level. Specifically, we redid the analysis of figure 2, but split the dataset by day of scanning. Stability was very high for LE and DM ($r > 0.99$) and moderate-to-high for PCA ($r \approx 0.72$) (Supplementary Figure1)

In our example (Figure 6), the correlations between FC gradients and T1w/T2w stay significant (two-tailed, $p < 0.001$) even when comparing the correlation to 1000 null models whereas correlations between FC gradients and cortical thickness was non-significant (two-tailed, $p = 0.12$).